# Caveolin-1 in Skin Protection Against Radiation-Induced Skin Injuries: Pathophysiological Mechanisms and New Avenues for Prevention

**DOI:** 10.3390/ijms27010415

**Published:** 2025-12-30

**Authors:** Ilja L. Kruglikov

**Affiliations:** Scientific Department, Wellcomet GmbH, 76646 Bruchsal, Germany; i.kruglikov@wellcomet.de

**Keywords:** radiation-induced skin injuries, caveolin-1, prevention, therapy

## Abstract

The identification of caveolin-1 (CAV1) as a universal pathophysiological factor and target for treating various cutaneous conditions and the recognition of its role as a universal factor and target in the protection of cells from genotoxic stress have opened new avenues for protecting skin against radiation-induced skin injuries (RISIs). A significant and rapid increase in CAV1 content in irradiated cells, reaching a maximum at 30–60 min after irradiation, coupled with internalization of epidermal growth factor receptors involved in the activation of homologous recombination and non-homologous end-joining repairing of double-strand breaks in affected cells, can protect the cells from irradiation to some degree. However, a higher level of protection can be achieved when the CAV1 content in the skin is increased before irradiation. Such an enhancement in the expression and translocation of CAV1 can be induced by the local application of thermo-mechanical stress with parameters inducing reinforcement of the actin cytoskeleton in treated cells. The application of very-high-frequency ultrasound waves with frequencies above 10 MHz or combined multi-frequency ultrasound waves can provide new means of protecting against RISIs during radiation therapy without reducing the radiosensitivity of cancer cells.

## 1. Introduction

Radiation-induced skin injuries (RISIs) are common complications of radiation therapy observed in 85–95% of treated patients, ranging from transient radiation dermatitis to chronic skin ulceration, radiation-induced keratosis, and fibrosis [1]. RISIs can lead to mild-to-severe skin reactions; develop over several days or weeks after therapy or appear as late reactions months or even years after irradiation; induce pain; and result in long-lasting and significant reduction in quality of life. Early efforts to counteract RISI appearance were directed toward improving radiation schedules to reduce the effective dose of radiation, in order to remain under the threshold dose for corresponding side effects. Later, attention was switched to the mechanisms responsible for RISIs [2] to improve their treatment and provide some degree of skin protection.

It is widely accepted that the most serious effects of ionizing radiation (IR) are related to the production of DNA damage in the form of double-strand breaks (DSBs). DSBs can lead to either premature cell senescence or cell death, accompanied by severe tissue reactions such as ulceration and necrosis. Irradiated tissue exhibits chronic inflammation with significant secretion of proinflammatory factors [3]. This process is connected to the activation of the complement system in the skin, leading to the production of anaphylatoxins C3a and C5a [4], which are the chemo-attractants for inflammatory immune cells. Additionally, IR induces premature senescence in keratinocytes, which adopt a special secretory phenotype and stimulate an inflammatory response in affected tissue [5]. Radiation-induced fibrosis was found to be related to enhanced TGF-ß expression in the skin, the level of which strongly correlates with fibrosis scoring [3], while the appearance of non-healing radiation ulcers is thought to be caused by long-term radiation-induced damage to the vascular endothelium [6].

The broad variety of RISIs, characterized by distinct pathophysiological pathways, makes their overall prevention nearly impossible. However, it has recently become clear that there are some pathophysiological factors that are universally involved in different inflammatory and hyperproliferative cutaneous conditions and even connect these conditions with aging-induced senescence, wound healing, and cutaneous fibrosis. Thus, all the RISIs mentioned above should involve at least similar pathophysiological pathways. One of them is related to cellular signaling processes regulated by caveolin-1 (CAV1).

CAV1 is the principal structural component of caveolae, characteristic Ω-shaped plasma membrane invaginations in lipid rafts forming nanodomains typically measuring 50–100 nm. They are present in different skin cells, including keratinocytes, fibroblasts, adipocytes, melanocytes, and macrophages. Caveolae play a key role in volume control, cell adhesion, various signal transduction processes, and endo- and exocytosis. CAV1 mechanistically interacts with different membrane receptors, including TGF-β, matrix metalloproteinases (MMPs), heat shock proteins, hyaluronan, and toll-like and glucocorticoid receptors. Through these interactions, CAV1 is causally involved in processes such as collagen production, cutaneous inflammation, distribution and retention of water, regulation of the extracellular matrix, endo- and exocytosis, and autophagy [7].

Here, we reconsider the pathophysiology of RISIs, connecting it to the behavior of CAV1 in irradiated tissue, and propose new avenues for their prevention and treatment.

## 2. Caveolin-1 as a Universal Pathophysiological Factor and Target in Pathological Cutaneous Conditions

CAV1 content is downregulated in the context of various inflammatory skin conditions, including psoriasis and acne [7,8]. Very low levels of CAV1 have been observed in types of psoriasis such as plaque, pustular, palmoplantar, and nail psoriasis but not in guttate or inverse psoriasis [9]. Low levels of CAV1 are also typical in acne vulgaris [10] but not in hidradenitis suppurativa, in which it is significantly overexpressed [11]. The pathogenesis of atopic dermatitis is strongly associated with the behavior of lipid rafts in the plasma membranes of keratinocytes, and it was shown that targeted disruption of cholesterol via application of methyl-ß-cyclodextrin produced skin conditions similar to atopic dermatitis, which is characterized by high levels of extracellular release of interleukins [12]. Remarkably, skin conditions characterized by low CAV1 expression can be ameliorated through targeted stimulation of CAV1 [8]. Thus, CAV1 is not just a pathophysiological factor but also a target for the treatment of these diseases. For example, applying glucocorticoids in therapeutic doses—a gold standard in therapy for inflammatory cutaneous conditions—stimulates expression of CAV1, whose levels correlate with improvements in the state of skin [13]. Similarly, applying salicylic acid significantly stimulated CAV1 expression, and an increase in the level of this protein strongly correlated with amelioration of acne lesions [14].

Low expression of CAV1 is also typical in fibroproliferative cutaneous diseases such as fibrosis, hypertrophic scars, and keloids [15]. Through mechanistic interaction between CAV1 and TGF-β receptors, caveolae can trap and internalize these receptors via CAV1-dependent endocytosis and thus effectively restrict collagen production in the tissue [16]. Decreased CAV1 expression in the tissue cannot support this regulation, as recently demonstrated in regard to Dupuytren’s disease [17], another fibroproliferative condition.

Whereas CAV1 expression is temporally decreased in burns [8,18], it is significantly increased in chronic wounds, especially at the edges [13,18]. Elevated CAV1 levels in these areas prevent keratinocytes from penetrating the wound and thus inhibit its closure [13]. In an approach that was theoretically in line with this information but therapeutically very unusual, application of the anti-diabetic drug mevastatin—which reduces CAV1 expression—led to a significant improvement in the healing of chronic wounds [19]. Like in chronic wounds, CAV1 content is substantially increased in senescent cells, which are common in aged skin, and it has been stated that reducing CAV1 expression can be considered as a target in anti-aging therapy [8,20]. These results are summarized in Figure 1 (see also Table 1 in [21]).

Inflammatory skin conditions (such as radiation dermatitis), radiation-induced fibrosis, and chronic ulcers are the most common RISIs. Based on the information above, there is strong evidence supporting the assumption that CAV1 must be substantially involved in the pathogenesis of all these injuries. Since CAV1 is currently considered not just an important pathophysiological factor in different skin conditions but also an effective target for their treatment [8,21,22], incorporating CAV1 involvement in research may open new avenues for the prevention and treatment of RISIs.

## 3. CAV1’s Role Protecting Cells Against Genotoxic Stress

CAV1 is not just an important pathophysiological factor in different cutaneous conditions; it is also substantially involved in cells’ protective reaction against IR. As mentioned above, the most serious effect of IR in irradiated cells is related to the production of DSBs in DNA. These lesions can be repaired through either homologous recombination (HR) or non-homologous end joining (NHEJ), whereas NHEJ is much more error-prone. Various authors have demonstrated that CAV1 expression transiently increases in cells exposed to IR in a dose- and time-dependent manner, and this increase correlates with the activation of cell repair. Notably, this effect is not radiation-specific, can be caused by different types of DNA damage, and was observed in different cells capable of expressing endogenous CAV1.

Comprehensive analysis of this phenomenon has revealed that enhanced expression of CAV1 protects cells from DNA damage through the modulation of both the HR and NHEJ pathways [23]. This enhancement is not induced by transcription of the CAV1 gene and thus must be realized post-transcriptionally. IR-induced translocation of CAV1 occurs concurrently with intracellular transport of epidermal growth factor receptor (EGFR), which interacts with DNA-dependent protein kinase and promotes its phosphorylation, helping to enhance DNA repair [23,24]. As demonstrated in multiple studies, EGFR antagonists sufficiently reduce cells’ ability to repair radiation-induced DNA-DSBs.

In terms of temporal behavior, after IR was applied, CAV1 levels reached a maximum shortly after irradiation, followed by a slow decay. For example, in human breast cancer cells, these values were about 350% at 0.5 h and about 250% at 2 h after IR at 5 Gy [23]. This process can significantly increase EGFR translocation, which can be realized through clathrin- or CAV1-dependent endocytosis. Whereas the clathrin-dependent pathway is the main mechanism for EGFR internalization under physiological conditions [25], for which the endpoint is EGFR degradation [26], CAV1-dependent endocytosis is the most significant when cells are exposed to oxidative stress. Oxidative stress induces hyperphosphorylation of CAV1, which is essential for the transport of EGFR in its active form to a perinuclear compartment [26] and thus makes this type of endocytosis a crucial step in the intracellular radioprotective cascade (Figure 2). Since this endocytosis is tightly linked to the expression of microtubule-associated protein dynamin-2 (DNM2) in cell membranes [27], for its consistent enhancement, the expression of DNM2 in irradiated cells must also be increased, as demonstrated in in vitro experiments [28].

Notably, CAV1 is highly expressed in the endothelium and mechanistically interacts with endothelial nitric oxide synthase (eNOS), serving as this compound’s negative regulator. Radiation-induced CAV1 phosphorylation may disrupt this binding, leading to aberrant eNOS activity and loss of endothelial barrier integrity, which can also contribute to the pathophysiology of RISIs. Additionally, CAV1 and reactive oxygen species (ROS), which typically appear after IR, exhibit a bi-directional interaction: CAV1 negatively regulates ROS production, while IR-induced ROS can directly influence CAV1 levels in cell membranes. This effect can be realized through the activation of tyrosine kinase c-Src, which induces phosphorylation of CAV1, or through proteasomal degradation of CAV1 [29].

Similar CAV1 behavior was observed after the UV irradiation of diverse types of cells. Repeated irradiation of melanocytes in vitro with low doses of UV-B radiation significantly increased CAV1 expression in these cells [30]. The extent of CAV1 accumulation in chronologically and photoinduced aging skin is high, and researchers have proposed that CAV1 can serve as a marker for cellular senescence as well as a target for senescence reversion [20,31]. Similarly to IR, UV radiation induces quick nuclear translocation of EGFR in human keratinocytes, with a 10-fold increase in the nucleus at 30 min post-UV irradiation [32]. Congenial protective behavior of CAV1, with its translocation into the nuclei, was also observed in cells exposed to non-radiative oxidative stress [33].

All these observations reveal that the activation and translocation of CAV1 from the plasma membrane to the perinuclear area of cells constitute as a general mechanism protecting them from genotoxic stress, allowing us to consider the CAV1-dependent translocation of EGFR as a universal mechanism for the protection of cells against both intrinsic and therapy-induced oxidative stress [34]. In support of this concept, silencing CAV1 [23] or preventing its translocation to the nucleus [33] resulted in reduced survival of irradiated cells. This phenomenon is critical in radiotherapy for cancer since such overexpression of CAV1 in tumor cells will make them more resistant to radiation [35].

It should also be noted that the EGFR–CAV1 axis is disturbed in senescent skin cells, exhibiting greatly increased CAV1 content [20] concurrently with significantly reduced expression of EGFR [36]. Additionally, significantly increased levels of CAV1 in these cells allow stabilization of the caveolae and thus reduce their detachment from plasma membranes through endocytosis. Therefore, the protective mechanism based on the CAV1–EGFR axis should be attenuated in older individuals, potentially explaining why these patients are often at higher risk for RISIs than younger individuals.

The absolute value of CAV1 in affected tissue depends on the state of the skin before IR and on the applied irradiation scheme, which can lead to different scenarios. For example, long-lasting increased expression of CAV1 in such tissue caused by induction of its radioprotective effect without substantial endocytosis can lead to the development of RISIs such as chronic ulcers, whereas reduced content of CAV1 in this tissue caused by involvement of this protein in intensive endocytosis of EGFR will create conditions favoring the appearance of radiation dermatitis and radiation-induced fibrosis. This notion will need to be verified in future research.

## 4. IR-Induced Activation of Complement Pathway and Its Relation to CAV1

IR sufficiently activates the complement pathway, and this activation is essential in radiotherapy. Indeed, application of dexamethasone (an effective inhibitor of the complement pathway) before radiotherapy significantly reduces its efficacy [4]. IR induces increased production of anaphylatoxins C3a and C5a [4], which are involved in different cutaneous and subcutaneous adverse effects, among others, in fibrosis [37]. As we discussed recently, the complement pathway is also substantially activated in wounded skin, where it is mainly directed against *S. aureus*, and its improper activation can lead to various fibroproliferative cutaneous conditions [38]. Moreover, inhibition of the complement pathway has been considered as a target for addressing non-healing wounds [39].

The activation of the complement system necessarily affects RISIs through its terminal cascade leading to complement-dependent cytotoxicity associated with the formation of membrane-attacking complexes (MACs or C5b-9). To counteract this cytotoxicity and thus avoid autolysis, cells possess several defense mechanisms, including the ability to express inhibitors (such as membrane cofactor protein, CD46; decay-accelerating factor, CD55; and protectin, CD59) blocking the early stages of MAC assembly in the plasma membrane, remove opsonins (such as C3b) from the cell surface through their being shed by the matrix metalloproteinase MMP14, and remove MACs from the plasma membrane through CAV1/DNM2-dependent endocytosis [27,37]. A detailed illustration of these processes was presented in our recent publications [27,37,38].

The behaviors of CD59 and MMP14 after irradiation have mainly been investigated in cancer cells because of the wide application of IR in this field. Very recently, it was indeed demonstrated that IR significantly upregulates the expression of all early blockers of MAC formation (CD44, CD56, and CD59) in different tumor cells in a time- and dose-dependent manner, counteracting the MAC-dependent autolysis of these cells and reducing their radiosensitivity [40]. IR also sufficiently increased expression of MMP14 in glioma cells (as well as in other cancer cell lines), and inhibition of MMP14 made these cells more sensitive to irradiation [41]. These results provide further evidence of the protective role of MMP14 in radiotherapy. Concerning the clear radioprotective role of CAV1 and the involvement of CAV1/DNM2-dependent endocytosis in the radiosensitivity described above, enhancement of all three main pathways of cellular protection against MAC-dependent cell autolysis reflects significant involvement of the terminal cascade in the radiosensitivity of irradiated cells. Notably, CD44, CD55, and CD59 were found to be significantly upregulated in UV-B-irradiated cells too [42], indicating the universality of this mechanism.

Collectively, these results indicate that the self-defense cellular mechanisms are likely involved in the generation of RISIs. As we argued before, aberrant activation of the complement system can lead to fibrosis [37], which has been associated with low expression of CAV1 in affected tissue [15]. Such reduced expression of CAV1 after IR can be caused by enhanced translocation of CAV1 from the plasma membrane accompanied by insufficient production of the new CAV1, which will increase its intracellular concentration but simultaneously decrease the CAV1 content in the plasma membrane. Chronic radiative dermatitis, which can appear after a long latent period of months to years, is likely caused by the inability of cells to restore CAV1 expression in irradiated tissue, producing the latent internal conditions for an inflammatory reaction [7,8]. In contrast, chronic radiative ulcers are likely caused by chronically increased local overexpression of CAV1 in irradiated tissue. This insight opens new avenues for preventive/regenerative treatment of RISIs through the targeted modulation of CAV1 before or after application of IR or UV.

As mentioned above, IR induces premature senescence in keratinocytes, which stimulates an inflammatory response in affected tissue [5], indicating senescence is involved in the development of RISIs. Indeed, both high overexpression and severe Cav-1 deficiency can induce premature senescence in different types of cells [20]. However, while elevated CAV1 signaling contributes to replicative senescence or premature senescence following cellular stress, CAV1 deficiency promotes senescence in resting cells primarily through mitochondrial dysfunction [43]. Excessive accumulation of senescent cells exhibiting a senescence-associated secretory phenotype (SASP) in tissue can induce chronic diseases and severe tissue disfunction [44]. CAV1 is substantially involved in both apoptotic and immune-mediated clearance of senescent cells, and its modulation by IR can significantly influence these processes.

The apparent contradiction between low and high levels of CAV1 in radiative fibrosis and chronic radiative ulcers, which often coexist in IR patients, can be explained by the fact that just the edges of a chronic ulcer, but not the whole surface, exhibit high levels of CAV1 expression caused by the high production of the stress hormone cortisol in this area [13,18]. Thus, it is highly likely that the CAV1 expression pattern in irradiated tissue is inhomogeneous, which can lead to the appearance of mosaic spatial areas with either increased or decreased CAV1 levels. Additionally, it may be supposed that such inhomogeneity can influence the burden and clearance of senescent cells in the affected tissue and thus shift its state toward chronic inflammation, fibrosis, or ulceration. This intriguing possibility should be thoroughly investigated in future research.

## 5. New Avenues for Prevention of RISIs

The universal role of CAV1 as a target for the treatment of various inflammatory and hyperproliferative conditions, wound healing, and cellular senescence and its involvement in protecting cells against genotoxic stress allow us to consider CAV1 modulation as an important target in skin protection as well as in the regenerative treatment of RISIs. Importantly, any applied prevention should provide effective protection against RISIs without lowering radiation exposure for the cancerous area, which is usually located beneath the skin.

The level of CAV1 in irradiated cells has a biphasic effect on endocytosis: whereas high levels of CAV1 stabilize the caveolae and thus reduce the probability of their detachments from the plasma membrane, very low levels of CAV1 make the caveolae mechanically unstable and thus also reduce endocytosis. Thus, for effective enhancement of CAV1/DIN2-dependent endocytosis, which should allow effective transport of EGFR in its active form the plasma membrane to the perinuclear cellular compartment, an optimum value of CAV1 in the plasma membranes of irradiated cells is required.

CAV1 levels in the plasma membrane can be modulated in several ways, including by drugs or the local application of thermo-mechanical stress. It was recently reported that oral administration of aspirin (a non-selective inhibitor of cyclooxygenase) before application of IR in a murine model delayed the onset of RISIs and reduced their severity [45]. Earlier, it was shown that aspirin protected cells against genotoxicity through activation of the HR repair pathway [46], significantly increased CAV1 production in a dose-dependent manner in murine skin ex vivo [47], and even accelerated senescence in human dermal fibroblasts [48]. These results support the vital role of CAV1 in protecting cells against genotoxic stress. However, systemic application of aspirin or other drugs inducing enhanced expression of CAV1 can also cause overexpression of CAV1 in cancer cells, leading to a corresponding reduction in their radiosensitivity.

Caveolae are mechanosensitive structures that are linked to the intracellular actin network, and the application of thermo-mechanical stress can cause either the stiffening or fluidization of this network [49], depending on applied parameters such as mechanical strain and strain rate [15,50]. Sufficient thermo-mechanical stress in tissue can be generated, e.g., by applying very-high-frequency ultrasound (VHFU) at frequencies above 10 MHz [51,52]. Whereas applying high-intensity ultrasound leads to a stable fluidization of the cytoskeleton, applying low intensities induces its initial fluidization followed by recovery with subsequent reinforcement [49,53]. At the same ultrasound intensity, higher ultrasound frequencies induce greater strain in the cells, and this effect is more efficient than strain modification caused by applying increased intensity at the same frequency [54]. Excessive stress fibers appearing due to cytoskeletal reinforcement increase tension at the plasma membrane accompanied by an increase in the Cav1 level in this area, whereas reducing tension through cytoskeletal fluidization has the opposite effect [55]. Thus, depending on its parameters, thermo-mechanical stress can induce either stimulation or suppression of CAV1 expression.

This information explains the pronounced anti-inflammatory and anti-hyperproliferative effect of VHFU, as observed in different studies [56,57,58]. The impact of VHFU can be further enhanced by quasi-simultaneously applying ultrasound waves at different frequencies, known as LDM waves, where the single ultrasound frequencies quickly alternate with a frequency of up to several thousand Hz, which forces the cells to react concurrently to all applied frequencies (see the illustration in [58,59]). These combined waves not only have an anti-inflammatory effect [58] but can also significantly reinforce the tissue, as confirmed in detail via sonoelastography [59]. It has been reported that LDM waves are highly clinically efficient in the treatment of radiation-induced ulcer [56], radiation-induced fibrosis [60], and different post-operative side-effects and complications [57,61].

Moreover, LDM waves can be applied for pain management in patients with RISI. The pain-relieving effect of these waves has been connected to the uncoupling of electrical and mechanical waves simultaneously produced in the axonal membrane during its excitation. This effect can be induced either through direct disturbance of the mechanical surface waves in the axonal membrane or through shifting of the thermodynamic state of this membrane from its phase transition point [62]. Both effects can be realized through local application of LDM waves to the affected skin area. Different examples of the clinical application of LDM waves for pain treatment are given in [62].

Based on everything mentioned above, we can propose a new way for protecting the skin from RISIs based on pretreating the skin covering the area targeted for treatment with radiotherapy. This pretreatment must be performed under conditions allowing for an increase in the production of CAV1 in the skin but not in the area under the skin that should be targeted by IR. Such in-depth, differentiated regulation of CAV1 can be realized through applying VHFU or LDM waves at frequences above 10 MHz or even 19 MHz, which have penetration depths in the body of less than 3 mm or 1.5 mm, respectively. For example, applying 19 MHz ultrasound mainly localizes the treatment effect in the skin, which thickness is normally less than 1.8 mm. Such in-depth differentiated modulation of CAV1 will allow protection of the skin without substantially changing the radiosensitivity of deeper-lying cancer cells. Application of this protective method can have limitations in regard to aged skin characterized by a disturbed EGFR–CAV1 axis. Comprehensive future research is required to determine the optimal treatment protection schedules for different radiotherapeutic indications.

Based on the remarkable similarity between CAV1-dependent protective mechanisms in cells irradiated by IR or UV, it can be strongly supposed that such a preventive strategy can also be applied for the treatment of patients with melasma. This interesting issue will be discussed elsewhere.

## 6. Conclusions

The identification of CAV1 as a universal pathophysiological factor and target for the treatment of various cutaneous pathological conditions, and the recognition of its role as a universal factor and target in the protection of cells from genotoxic stress open new avenues for protecting skin against RISIs. A significant and quick increase in CAV1 content in irradiated cells, coupled with internalization of EGFR involved in the activation of repairing of double-strand breaks in affected cells, can—to some degree—protect the cells from irradiation. However, a higher level of such protection can be achieved when the CAV1 content in the skin is increased before irradiation. Such an enhancement of CAV1’s expression and its translocation to the plasma membrane can be induced by local application of thermo-mechanical stress using parameters inducing reinforcement of the actin cytoskeleton in treated cells. Applying very-high-frequency ultrasound waves with frequencies above 10 MHz or combined multi-frequency LDM waves can open new avenues in protection against RISIs during radiation therapy without reducing the radiosensitivity of cancer cells. This method can also be effectively applied for the regenerative treatment of existing RISIs through corresponding normalization of CAV1 content in affected skin areas.

## Figures and Tables

**Figure 1 ijms-27-00415-f001:**
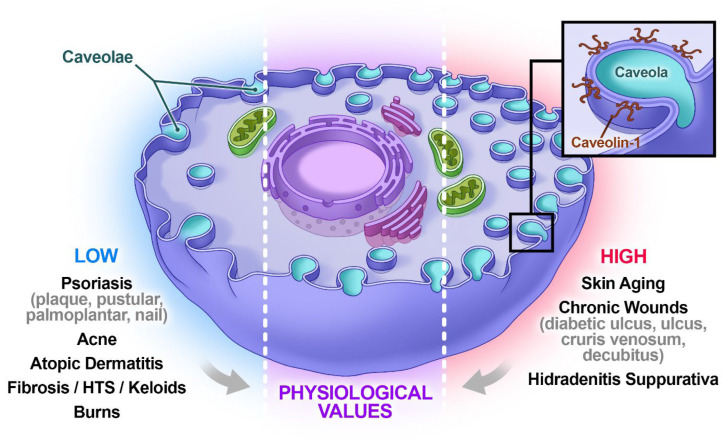
The involvement of CAV1 in different pathological cutaneous conditions. Among others, CAV1 levels are significantly reduced in the context of cutaneous conditions such as psoriasis, acne, atopic dermatitis, fibrosis/hypertrophic scars keloids, and burns, and it is strongly overexpressed in skin aging, chronic wounds, and hidradenitis suppurativa. These deviations from the physiological levels of CAV1 in plasma membranes deteriorate cell signaling and are also typical in corresponding RISIs.

**Figure 2 ijms-27-00415-f002:**
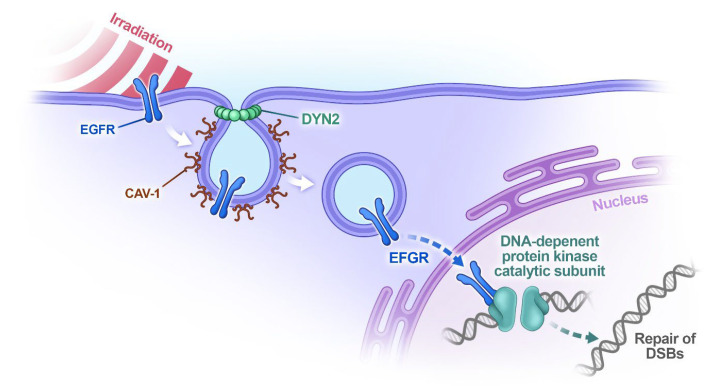
The role of the CAV1–EGFR axis in the protection of cells against IR. Ionizing radiation induces translocation of EGFR from the plasma membrane to the perinuclear space to induce DNA repair. This translocation is realized through CAV1-dependent endocytosis, which is essential for EGFR transport in its active form. This process sufficiently influences the CAV1 level in the plasma membrane and can lead to different scenarios, ranging from radiation dermatitis and radiation-induced fibrosis to chronic ulcers.

## Data Availability

No original data were collected or presented in this article.

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
