# Peer review of "Caveolin-1 in Skin Protection Against Radiation-Induced Skin Injuries: Pathophysiological Mechanisms and New Avenues for Prevention"

_ijms, 2025, doi:10.3390/ijms27010415_

Round 1
Reviewer 1 Report
Comments and Suggestions for Authors
1. While Section 4 notes vascular endothelial damage in chronic ulcers, the molecular mechanism is underdeveloped. CAV1, highly expressed in endothelium, binds and inhibits endothelial nitric oxide synthase (eNOS). Radiation-induced CAV1 phosphorylation may disrupt this binding, causing aberrant eNOS activity, impaired nitric oxide signaling, and loss of endothelial barrier integrity—key to RISI pathophysiology. Introduce a paragraph in Section 4​ discussing how radiation, via CAV1 phosphorylation, may dysregulate eNOS, thereby exacerbating endothelial dysfunction and inflammation. This addition would significantly strengthen the vascular pathology discussion.
2. Section 3 excellently covers CAV1's role in genotoxic stress/DNA repair. However, the critical role of radiation-induced reactive oxygen species (ROS) deserves more emphasis. A bidirectional relationship exists: CAV1 dysfunction increases oxidative stress, and conversely, ROS can modulate CAV1 expression/phosphorylation, creating a feedback loop. Expand Section 3​ to briefly explore this bidirectional regulation. For instance, how might ROS modify CAV1, and how does CAV1 influence antioxidant responses? This would solidify CAV1's role as an oxidative stress node.
3. The manuscript correctly identifies CAV1 as a senescence marker. Yet, its potential role in RISI chronicity via affecting senescent cell clearance is unexplored. CAV1 deficiency could impair immune-mediated clearance of senescent cells, leading to their accumulation and persistent SASP-driven chronic inflammation/fibrosis. Enhance the discussion of chronic RISI (e.g., in Section 4)​ by proposing a hypothesis: Could the spatial heterogeneity of CAV1 expression after irradiation influence local senescent cell burden and clearance, thereby steering tissue toward fibrosis or ulceration? This offers a novel perspective on long-term RISI.
Author Response
Thank you for your professional and constructive comments. This is my pleasure to work with Reviewer from the field.Please consider that, according to your comments, I added three new references 29, 43, 44; the numeration of other references was correspondingly shifted. Editing service by IJMS was provided in accordance with recommendations of Rev #2. This prohibited the use of full text tracking. All changes in the text were marked in red.To better understand my answers, please also consider the following. This article was written not as a comprehensive Review but as a Prospective. The main idea – different common radiation-induced skin injuries include the same important pathophysiological step connected to modulation of CAV1 content in affected tissue. Considering the important role of CAV1 in skin protection against IR, this naturally leads to assumption that a proper modulation of CAV1 before IR application or after appearance of RISI can prevent/reduce their development. In this article, I wanted to connect the dots and show the new ways for future research.
Comment #1: While Section 4 notes vascular endothelial damage in chronic ulcers, the molecular mechanism is underdeveloped. CAV1, highly expressed in endothelium, binds and inhibits endothelial nitric oxide synthase (eNOS). Radiation-induced CAV1 phosphorylation may disrupt this binding, causing aberrant eNOS activity, impaired nitric oxide signaling, and loss of endothelial barrier integrity—key to RISI pathophysiology. Introduce a paragraph in Section 4 discussing how radiation, via CAV1 phosphorylation, may dysregulate eNOS, thereby exacerbating endothelial dysfunction and inflammation. This addition would significantly strengthen the vascular pathology discussion.
Thank you for this comment. You address the question which I also considered as I prepared this article. Yes, CAV1 phosphorylation may induce aberrant eNOS activity and this process can be considered as one of the pathophysiological steps in chronic ulcers. However, I have mentioned in Section 1 the “vascular endothelial damage” just as an example (together with TGF-ß for fibrosis or activation of anaphylatoxins in inflammation) to demonstrate that in current literature the pathophysiologies of various RISI are considered as fully different. It was needed to make the main statement in this section - “Such a broad variety of RISI characterized by distinct pathophysiological pathways makes their general prevention hardly possible”. I strongly believe that the “vascular endothelial damage” is not the main factor for development of chronic ulcers and not a target for their treatment. Several years ago, the group of Ivan Jozic (Miami, USA) started (parallel to us) to make a systematic cataloging of CAV1 behavior in different pathological cutaneous conditions, among many others in burns and chronic wounds. According to their well-documented results, the main problem in chronic wounds appears to be the highly elevated level of CAV1 at the wound edges, which prohibits entering of keratinocytes into the wound. The reason for this is a high expression of cortisol in this area, which level is directly connected to CAV1. Two of their articles concerning this issue were referred in my article as [13,18], but were not discussed in depth.That means, the vascular endothelial damage is surely present in chronic wounds, but it cannot be the target. Concentration on this issue will deviate the reader from the main topic – CAV1 is a pathophysiological factor and a target in all common RISIs, and its proper modulation can sufficiently influence the outcome of these severe side effects.According to your recommendation, I added a sentence in section 3 describing the connection between CAV1 phosphorylation and eNOS but did not deeply discuss this issue.
Comment #2: Section 3 excellently covers CAV1's role in genotoxic stress/DNA repair. However, the critical role of radiation-induced reactive oxygen species (ROS) deserves more emphasis. A bidirectional relationship exists: CAV1 dysfunction increases oxidative stress, and conversely, ROS can modulate CAV1 expression/phosphorylation, creating a feedback loop. Expand Section 3 to briefly explore this bidirectional regulation. For instance, how might ROS modify CAV1, and how does CAV1 influence antioxidant responses? This would solidify CAV1's role as an oxidative stress node.
I added the following text to this section: “Additionally, CAV1 and reactive oxygen species (ROS) which typically appear after IR demonstrate a bi-directional interaction: CAV1 negatively regulates ROS production, while IR-induced ROS can directly influence CAV1 level in cell membranes. This effect may be realized through activation of tyrosine kinase c-Src inducing phosphorylation of CAV1 or through proteasomal degradation of CAV1 [29]“. Here a new reference #29 was added.
Comment #3: The manuscript correctly identifies CAV1 as a senescence marker. Yet, its potential role in RISI chronicity via affecting senescent cell clearance is unexplored. CAV1 deficiency could impair immune-mediated clearance of senescent cells, leading to their accumulation and persistent SASP-driven chronic inflammation/fibrosis. Enhance the discussion of chronic RISI (e.g., in Section 4) by proposing a hypothesis: Could the spatial heterogeneity of CAV1 expression after irradiation influence local senescent cell burden and clearance, thereby steering tissue toward fibrosis or ulceration? This offers a novel perspective on long-term RISI.
This is surely correct. But I had a reason why I did not discuss this issue in the original version of the article. There are two diametrically opposed groups of cutaneous pathologies connected to CAV1 expression - those with under-expression and with over-expression of CAV1 (they were illustrated in Fig. 1). As we discussed earlier (PMID: 31493519), “severe Cav-1 deficiency can also produce premature senescence in at least some cell types, and this effect was connected with associated mitochondrial dysfunction ... Therefore, both overexpression and strong suppression of Cav-1 levels can lead to premature cellular senescence …”. For example, psoriasis is an inflammatory condition characterized by very low level of CAV1 expression but with upper keratinocytes exhibiting a senescent-like phenotype. SASP contributes to elderly atopic dermatitis, whereas AD lesions generally demonstrate low level of CAV1. The same is correct for keloids, which have different features of senescence but also a low CAV1 expression. I did not want to start any discussion concerning such biphasic behavior of CAV1 in senescence since it should consequently lead to discussion of the role of senescence in inflammatory and fibroproliferative skin conditions. I think, this article is wrong place for such discussion.
However, I acknowledged your proposal and added the following text to the Section 4: “As we mentioned above, IR induces premature senescence in keratinocytes, which stimulate an inflammatory response in affected tissue [5] and points to involvement of senescence in development of RISI. Indeed, both strong overexpression and severe Cav-1 deficiency can induce premature senescence in different types of cells [20]. However, while elevated CAV1 signaling contributes to replicative senescence or premature senescence following cellular stress, CAV1 deficiency promotes senescence in resting cells primarily through mitochondrial dysfunction [43]. Excessive accumulation of senescent cells demonstrating senescence-associated secretory phenotype (SASP) in the tissue can lead to development of chronic diseases and severe tissue disfunction [44]. CAV1 is substantially involved in both apoptotic and immune-mediated clearance of senescent cells, and its modulation by IR can significantly influence these processes… Additionally, it may be supposed that such inhomogeneity can influence the burden and clearance of senescent cells in affected tissue and thus shift its state toward chronic inflammation, fibrosis or ulceration.“ Here, it was needed to add two new references #43 and #44.
Reviewer 2 Report
Comments and Suggestions for Authors
Dear Authors
This manuscript has a novetly regarding caveolin-1 against radiation-induced skin injuries"
However, there were so many shortcomings and errors.
Here are major comments.
1. Many sentences were too long and complicated. Thus, an English editing service is neeeded.
2. Please, strengthen the Introduction part, including the main subject, caveolin-1
3. Please, provide a few Tables regarding caveolin-1 in pathological cutaneous conditions and protection of cells against genotoxic stress.
4. Figure 1 is not enough to emphasize your message. Please, provide more figures.
Comments on the Quality of English LanguageMany sentences are overly long and complex, making them difficult to read. In addition, some expressions are awkward or unnatural. English editing service should be needed.
ex) Identification of caveolin-1 (CAV1) as a universal pathophysiological factor and target for treatment of various cutaneous conditions as well as recognition of its role as a universal factor and target in protection of cells from genotoxic stress opens new avenues for skin protection against radiation-induced skin injuries (RISI). Significant and quick increase of CAV1 content in irradiated cells, reaching its maximum at 30-60 minutes after irradiation and coupled with internalization of epidermal growth factor receptors involved in activation of homologous recombination and non-homologous end joining repairing of double strand breaks in affected cells, can to some degree protect the cells from irradiation.
Author Response
Thank you for your comments. Please consider that, according to recommendations of Rev #1, I added three new references 29, 43, 44; the numeration of other references was correspondingly shifted. Editing service by IJMS was provided in accordance with your recommendation. This prohibited the use of full text tracking. All changes in the text were marked in red.
Comment #1: Many sentences were too long and complicated. Thus, an English editing service is needed.
Response #1: I followed your proposal. The text was edited by editorial service of IJMS.
Comment #2: Please, strengthen the Introduction part, including the main subject, caveolin-1
Response #2: You are right. This was done, the first paragraph of the text from Section 2 was shifted to Section 1 “Introduction”.
Comment #3: Please, provide a few Tables regarding caveolin-1 in pathological cutaneous conditions and protection of cells against genotoxic stress.
Response #3: I respectfully disagree. This is not the comprehensive Review but a Perspective article. All needed information concerning the CAV1 in different pathological conditions was described in the text with proper references. Additional information including corresponding Tables can be found in our former publications in Aging Res Rev, npj Aging Mech Dis, npj Regenerative Medicine, Exp Dermatol, and Int J Mol Sci and in ref. 21, 22, as it was referenced in the text. I don’t see any need to repeat this information in present article, especially since this is not its main topic.
Comment #4: Figure 1 is not enough to emphasize your message. Please, provide more figures.
Response #4: I did not understand your concern, since there are two Figures in the text of the article.The main idea of this article – different common radiation-induced skin injuries share the same important pathophysiological step connected to modification of the CAV1 content in affected tissue. Considering the important role of CAV1 in skin protection against IR, this naturally leads to assumption that a proper modulation of CAV1 before IR application or after appearance of RISI can prevent/reduce their development. In this article, I wanted to connect the dots and show the new ways for future research. Correspondingly, Figure 1 ilustrates the behavior of CAV1 in different pathological skin conditions, among others, in fibrosis, dermatitis, and chronic wound, which are the most common RISI. All these conditions are characterized by sufficient deviation of the CAV1 level from its physiological value. To make it clearer, the Legend to Figure 1 was changed. Figure 2 demonstrates the involvement of CAV1 in protection of cells against ionizing radiation. Here I also changed the Legend to make it better understandable for the readers. Moreover, in Section 4 I referred to additional illustrations given in our articles published in Nat Rev Endocrinol, Obesity and Front Immunol. I don’t think this is a good style to copy the Figures from other (even own) articles or just to slightly modify them.
Round 2
Reviewer 1 Report
Comments and Suggestions for Authors
can be accepted
Author Response
Thank you.
Reviewer 2 Report
Comments and Suggestions for Authors
Dear Author
1. Although you mentioned the IJMS editing service, there are still so many long subjects in many sentences.
To improve readability for readers, the sentence was revised by reducing the length of the subject. After revising this manuscript as a second major revision, you should submit the editing service document as non-publishable material.
2. Despite the request for tables, a response that primarily directs readers to previously published references may diminish the intended value of this narrative review.
For even Narrative Review articles in IJMS, the inclusion of summary tables to organize key concepts is strongly favored.
3. Figures 2 are okay. However, please adjust the placement of the figures to make sure they are close to the relevant sentences.
4. Please, add more references of at least 90~100. The author should consider the quality of the recent IJMS.
Comments on the Quality of English LanguageMany sentences are overly long and complex, making them difficult to read. In addition, some expressions are awkward or unnatural. English editing service should be needed.
ex) Identification of caveolin-1 (CAV1) as a universal pathophysiological factor and target for treatment of various cutaneous conditions as well as recognition of its role as a universal factor and target in protection of cells from genotoxic stress opens new avenues for skin protection against radiation-induced skin injuries (RISI). Significant and quick increase of CAV1 content in irradiated cells, reaching its maximum at 30-60 minutes after irradiation and coupled with internalization of epidermal growth factor receptors involved in activation of homologous recombination and non-homologous end joining repairing of double strand breaks in affected cells, can to some degree protect the cells from irradiation.
Author Response
Dear Reviewer,
Thank you for your comments. Here is my point-by-point answer.
Comment 1: Although you mentioned the IJMS editing service, there are still so many long subjects in many sentences. To improve readability for readers, the sentence was revised by reducing the length of the subject. After revising this manuscript as a second major revision, you should submit the editing service document as non-publishable material.
Response 1: Unfortunately, this is a very strange situation, which I never faced before. In accordance with your proposal, I applied for the Editing service of IJMS. Now you demand to change the style of writing. Please consider that I have more than 200 scientific publications (many of them in very high IF journals ) and believe to know, how to write the scientific article. I don't think this style is a problem for the readers, since the majority of these articles are highly cited.
Comment 2: Despite the request for tables, a response that primarily directs readers to previously published references may diminish the intended value of this narrative review. For even Narrative Review articles in IJMS, the inclusion of summary tables to organize key concepts is strongly favored.
Response 2: I understand your concern, however, there are two problems with this. During the initial phase of article application, there was a discussion with Editorial Office about this issue. We agreed that this article is not the Comprehensive or Narrative Review but the Prospective article. Also, Editorial Office of IJMS asked me "to add two Figures / Tables". I decided me for Figures, which were produced by professional scientific illustration artist. Now you imperatively demand to additionally input the Table(s) summarizing the discussion presented in the text. I think you don't know about the discussion with Editorial Office, but you will surely understand that I cannot make the modifications of the text ad infinitum. Additionally, as I mentioned in the first Rebuttal, such summarizing was provided in several articles, among others Table 1 in the Ref 22. To emphasize this, I slightly changed the text as "These results are summarized in Figure 1 (see also Table 1 in [22])". I also make the reviews for very high impact journals and consider the re-publishing of the Tables from one article to another as a bad style and know that many collugues have the same opinion. This is the reason why I cannot fully follow your suggestion to add the Table(s).
Comment 3: Figures 2 are okay. However, please adjust the placement of the figures to make sure they are close to the relevant sentences.
Response 3: The figures are mentioned on the right places: Fig.1 - after description of CAV1 behavior in various pathological skin conditions, Fig.2 - after discussion of the mechanistic interaction between CAV1 and EGFR. However, this is the decision of the Editorial Office where the Figures should be placed in the printed version (when accepted).
Comment 4: Please, add more references of at least 90~100. The author should consider the quality of the recent IJMS.
Response 4: First, I don't beleive that the quality of the article is somehow connected to the number of references. Much more it is connected to the scientic content and the novelty, which you did not mention in your comments. Second, as I mentioned before, we have discussed this issue with Editorial Office during the first step of the editorial process, since Editorial Office also first considered this article as a Review and proposed to increase the number of references. In my mail to the Editorial Office written on the 2025/11/27, I explained why it is not possible. I have written: "Generally, this article is not a Comprehensive Review but a Prospective one. It can be also considered as a Viewpoint, however, there is no such article type by IJMS. It contains the latest research in the field, not much more was done till now... Please let me know whether it is possible to consider this article as a Prospective or a Viewpoint. If this is not possible, I must consider another possibility for publication. For me this would be not the best solution since I had a good experience with your journal before." My arguments were accepted by the Editor and that was the main reason for me to continue.
You will surely understand that I cannot start to change the whole article just because one Reviewer does not know or does not want to accept the aggreement between the author and the journal.